# EC-YOLO: Improved YOLOv7 Model for PCB Electronic Component Detection

**DOI:** 10.3390/s24134363

**Published:** 2024-07-05

**Authors:** Shiyi Luo, Fang Wan, Guangbo Lei, Li Xu, Zhiwei Ye, Wei Liu, Wen Zhou, Chengzhi Xu

**Affiliations:** School of Computer Science, Hubei University of Technology, Wuhan 430068, China; 102201058@hbut.edu.cn (S.L.); 20021026@hbut.edu.cn (F.W.); 20000012@hbut.edu.cn (G.L.); hgcsyzw@hbut.edu.cn (Z.Y.); 20071014@hbut.edu.cn (W.L.); zw_mmwh@hbut.edu.cn (W.Z.); xcz911@hbut.edu.cn (C.X.)

**Keywords:** deep learning, electronic components, PCB, object detection

## Abstract

Electronic components are the main components of PCBs (printed circuit boards), so the detection and classification of ECs (electronic components) is an important aspect of recycling used PCBs. However, due to the variety and quantity of ECs, traditional target detection methods for EC classification still have problems such as slow detection speed and low performance, and the accuracy of the detection needs to be improved. To overcome these limitations, this study proposes an enhanced YOLO (you only look once) network (EC-YOLOv7) for detecting EC targets. The network uses ACmix (a mixed model that enjoys the benefits of both self-attention and convolution) as a substitute for the 3 × 3 convolutional modules in the E-ELAN (Extended ELAN) architecture and implements branch links and 1 × 1 convolutional arrays between the ACmix modules to improve the speed of feature retrieval and network inference. Furthermore, the ResNet-ACmix module is engineered to prevent the leakage of function data and to minimise calculation time. Subsequently, the SPPCSPS (spatial pyramid pooling connected spatial pyramid convolution) block has been improved by replacing the serial channels with concurrent channels, which improves the fusion speed of the image features. To effectively capture spatial information and improve detection accuracy, the DyHead (the dynamic head) is utilised to enhance the model’s size, mission, and sense of space, which effectively captures spatial information and improves the detection accuracy. A new bounding-box loss regression method, the WIoU-Soft-NMS method, is finally suggested to facilitate prediction regression and improve the localisation accuracy. The experimental results demonstrate that the enhanced YOLOv7 net surpasses the initial YOLOv7 model and other common EC detection methods. The proposed EC-YOLOv7 network reaches a mean accuracy (mAP@0.5) of 94.4% on the PCB dataset and exhibits higher FPS compared to the original YOLOv7 model. In conclusion, it can significantly enhance high-density EC target recognition.

## 1. Introduction

With the rapid growth of scientific and technological development, the number of electronic products is increasing rapidly and the lifespan of electronics products is decreasing, resulting in the generation of large amounts of e-waste. PCBs (printed circuit boards) are extensively used in electronic and electrical products and make up almost 3–6% of all e-waste of all types and sizes, which makes them among the fastest growing waste streams. ECs (electronic components) are the main components of PCBs, and although many waste ECs are still in service, the PCBs have lost their overall functionality. Because the common life of an EC far exceeds its design life, most waste EC can still be recycled for secondary use. In addition, in terms of resource recovery, some waste ECs have nearly all of the resources of rare metals that make up PCBs, and the recycling values of some ECs even surpass the recycling values of waste PCBs. However, ECs contain numerous toxins, and their compositions are very complex. For this reason, an important complement to the recycling of used PCBs is the efficient and environmentally sound treatment of used ECs. After disassembly from PCBs, ECs often consist of a combination of capacitors, resistors, and inductors. Efficient identification and classification of ECs is required to reuse and recover them due to the different functions and compositions of different types of ECs.

Some traditional recycling of discarded PCBs was initially reported to be done by human operators melting solder and manually removing ECs. The use of manual disassembly in large-scale industrial recycling has been hindered by low processing capacity and high health risks to the workforce [1]. As a result, labour-saving methods to remove ECs from bare boards have been developed, such as mechanised recycling [2], heat desoldering, and chemical reagents [3]. Through an automated disassembly process, ECs are freed from the bare boards. However, relevant studies on categorizing hybrid ECs into specific categories are still lacking.

In recent years, the development of EC non-contact detection technologies using deep learning methods has been significant. There are various types of ECs with differing shapes. The target detection technology simulates the principle of visual cognition in the brain, and by extracting the features of the retained objects, the objects can be recognized even if they reappear at different scales, directions, and positions. Therefore, the combination of detection of ECs with the detection of target molecules is necessary. In the last few years, the successful application of computer vision with mature deep learning techniques for automatic EC classification has brought great progress for further accurate recovery of waste resources. These techniques include convolutional neural networks (CNNs [4]), region-based CNN (R-CNN [5]), multilayer perceptron (MLP [6]), and support vector machine (SVM [7]). Nowakowski et al. [8] used CNN (a deep learning convolutional neural network) to identify and categorise specific types of electronic waste. Liang and Gu et al. [9] proposed a multitask learning model for simultaneous localization and identification of waste materials. A graph convolutional network was proposed to detect components on printed circuit boards by Kuo et al. [10]. It is worth noting that their application in waste classification is limited due to the relatively slow computational power and time delay. There is a need to apply more advanced methods for small-sized waste ECs for fast identification and online sorting. The YOLO [11] (you only look once) network transforms the problem of classification and localization of the target into a regression problem, which results in improved speed of detection compared to other CNNs. In such work, for the detection of SMDs (small surface-mounted devices on printed circuit boards), Li et al. [12] proposed an enhanced YOLOv3 network. To address the problem of automatic disassembly and recycling of electronic components on printed circuit boards, Chen et al. [13] proposed a combined algorithm based on YOLOv5 and hierarchical classification algorithms. Our work attempts to investigate better YOLO networks that focus on the detection and classification of different types of ECs to accurately recycle these materials.

In this paper, to address the challenges of recognizing and classifying ECs when they are recovered, we propose EC-YOLOv7, which is an advanced and efficient network model for EC detection. The model proposed is an improvement of the YOLOv7 architecture that can more precisely detect densely packed and diverse micro ECs in the intricate background of the image. The new method should overcome the limitations of earlier studies and provide better detection performance in different light, colour, and state detection situations. These improvements contribute to the high efficiency of EC recovery. Our work’s primary contributions are as follows.

Aiming at the multi-scale and large-scale variability of ECs, we design a DyHead detection module that can provide size, task, and spatial awareness simultaneously. This design enhances both small and large target acquisition, strengthens the processing of complex scenes and multi-targets, and improves the algorithm performance without adding additional computational complexity.A new AC-E-ELAN module is intended to replace E-ELAN, and the ResNet-ACmix module is deployed in the backbone. The optimized backbone network enhances target detection performance, improves object representation and understanding, and speeds up training and inference.The SPPCSPC module in the neck is improved to be an SPPFCSPC module to enhance the speed of fusing image features. This module improves the robustness and recognition accuracy of the model for complex EC targets.A soft NMS using the WIoU loss function is proposed to address the target overlap problem and improve the small target detection accuracy. Compared to the original NMS, Soft-NMS is better equipped to handle overlapping bounding boxes, reduce duplicate detections, and improve accuracy by adding a WIoU penalty term.

The rest of the paper is structured as follows. Section 2 describes the source of the innovative ideas in this paper. Section 3 details the methodology for model improvement. Section 4 describes the experimental design of the paper and analyses the results. Section 5 summarizes the paper.

## 2. Related Work

In this age of Industry 4.0, more and more intelligent ways are being used to dispose of PCBs [14]. The IoT uses deep learning classification and cloud computing technology to accurately classify waste at the start of collection, reducing the costs of waste classification, monitoring, and collection [15]. EC components vary widely in type, appearance, and size. For this reason, conventional inspection methods that rely on human vision are ineffective and prone to error. Therefore, automated production requirements cannot be met by traditional methods. Recently, sophisticated deep learning techniques from computer vision have been used to classify waste at an early stage. Vision-based inspection can be used to detect and classify electronic components on PCBs in electronics manufacturing. Using convolutional neural networks (CNNs), Atik [16] proposed a new model for classifying ECs. Hu et al. [17] proposed an NH-CNN-based algorithm for EC detection and classification to reduce computational complexity. Xu et al. [18] demonstrated an EC recognition algorithm that uses a Faster-SqueezeNet network to reduce the size of the parameters of the network and the complexity of the computations. In addition, various researchers have used well-known architectures such as AlexNet [19], VGGNet [20], GoogLeNet [19], InceptionV3 [21], ResNet [22], and DenseNet [23] for EC recognition. However, their applications in waste classification are limited due to the relatively slow computational power and time delay. It is worth noting that the classification of ECs has been an industry challenge due to the variety and quantity of ECs. There is a need to apply more advanced methods for small-sized waste ECs for fast identification and online sorting.

The techniques mentioned above have proven to be effective for waste classification. However, it is important to note that they are restricted to the classification of e-waste images. As previously stated, deep neural networks perform poorly at identifying and recognizing e-waste in PCB images. This is due to the complex background of a PCB’s electronic component images and the small size and large number of ECs. YOLO is an advanced target detection method because it can deal with images in real time and is better at generalising the target representation than other deep learning models. The YOLO network improves detection speed over other CNNs by transforming the problem of target classification and localisation into a regression problem. Wahyutama et al. [24] investigated a method that uses a webcam and real-time object detection with YOLO to separate and collect recyclables from trash cans. Huang et al. [25] proposed a YOLOv2-based resistive capacitor detection and classification method. Li et al. [26] developed a real-time network for EC detection using YOLOv3 to match the actual sensor field size and the anchor size. Zhang [27] used deep learning YOLO to identify EC components in 450 images and 10,500 PCB samples, and the model can detect PCBs with different lighting, colour backplanes, and material numbers with 98% accuracy. Du et al. [28] proposed an enhanced YOLOv5s network called YOLO-MBBi for detecting surface defects on PCBs to address the deficiencies of existing methods for detecting PCB surface defects. Ling et al. [29] proposed an improved YOLOv8 deep learning model for efficient and accurate detection of densely arranged EC elements. Based on the aforementioned research, we conclude that improving the network layout and architecture of the YOLO family of algorithms is a key direction of development to achieve better performance in practical EC detection application scenarios.

Although major advances have occurred in EC detection using advanced image processing and machine learning techniques, much remains to be done. These YOLOs use a large number of parameters, which makes them slow at recognition, especially at high resolutions. However, embedding them into real industrial inspection equipment can be difficult because the use of a high number of parameters can result in heavyweight training models. Furthermore, there is a need to enhance the accuracy of detecting small targets in previous research to meet industry standards. Therefore, it is imminent to consider a new lightweight end-to-end model for EC detection. In this paper, YOLOv7 is improved to address the shortcomings of YOLOv7 in this area.

## 3. Methods

### 3.1. YOLOv7

The YOLOv7 model architecture is a derivation of the YOLOv4, scaled YOLOv4, and YOLO-R models. The pre-processing strategy of the YOLOv7 model is combined with the pre-processing technique of the YOLOv5 model, and the technique of mosaic data enhancement is used to detect small targets. As an architectural extension of ELAN, E-ELAN (Extended ELAN) is proposed. The computational block that serves as the backbone of YOLOv7 is known as E-ELAN. Extend, filter, and merge bases continually improve the network’s ability to learn without affecting the gradient path. Group convolution is used to increase the channels and bases of the arithmetic blocks in the architecture of the arithmetic blocks. Different features are acquired by instructing various sets of computational blocks. For connection-based models, YOLOv7 introduces composite model scaling. In order to obtain the best structure, the composite scaling approach preserves the initial properties of the model. The model is focused on a small number of trainable optimising modules and a “BoF [30]” (bundle of free) technique. BoF is a strategy for improving model performance without increasing training costs. The network structure of YOLOv7 is illustrated in Figure 1 and comprises the following main components:Input: The first step in YOLOv7 is to enhance the mosaics by randomly slicing, scaling, and joining the data to add targets to the dataset [31]. Then, optimal anchor points are adaptively computed using the training set. Subsequently, the images undergo resizing to reach a normalized size before being passed to the trunk.Trunk: The trunk’s primary function is to extract features, and it utilizes CBS, ELAN, and MP structures. The CBS architecture comprises a convolutional (Conv) layer, a batch normalization (BN [32]) layer [31], and a SiLU (sigmoid-weighted linear unit). CBS is primarily used for image channel transformation, feature extraction, and image subsampling. ELAN consists of two major steps: a 1 × 1 convolution to obtain image features and four 3 × 3 convolution sections to obtain further image features. By extracting complementary features and optimising gradient paths, the method improves the robustness of the target detection network. In order to achieve multi-scale target detection, the MP module (multi-prediction module) is used. The system is able to efficiently detect objects with different sizes by means of feature extraction and multi-stage predictive analysis.Neck: The section on the neck comprises three output channels for feature mapping. The SPPCSPC [33] module comprises of two components: SPP (spatial pyramid pooling) and CSPC (channel spatial pyramid pooling). The neck network is a bi-directional fusion backbone network that operates in both a top-down and bottom-up manner. It achieves multiscale network fusion by collapsing features between the backbone and detection layers.Head: The network of heads comprises three distinct dimensions of detected heads. The RepVGG [34] module is used to predict the outcome of the detection task to improve the overall performance of the model.

### 3.2. Dynamic Head Module

The necking network of YOLOv7 bridges the bottom and top features by constructing a multilevel feature pyramid to achieve efficient detection and identification of targets at multiple scales. However, given the variety and size diversity of electronic components (ECs) on printed circuit boards (PCBs), traditional necking network structures have limitations in coping with such complex scale variations and fine resolution of spatial information, which affects their effectiveness at detecting ECs. To overcome this challenge, we integrate the DyHead (dynamic head [35]) structure with the basic YOLOv7 model to enhance the model’s adaptability and performance at EC detection tasks. DyHead achieves deep focusing and scale adaptation enhancement of target features by integrating the target detection head with advanced attention mechanisms. As shown in Figure 2, DyHead not only optimises the structure of the model and significantly improves the detection accuracy but also achieves efficient use of computational resources. The dynamic characteristics of DyHead enhance the model’s ability to perceive EC size changes in complex PCB scenes, ensuring robust performance for diverse electronic component detection, especially when dealing with tiny components and complex layout structures; it demonstrates excellent detection performance and spatial information capturing ability.

Figure 2a demonstrates the conversion of the feature map into a 3D tensor. The tensor has L layers, C channels, and S height and breadth of the signature map. As shown in Figure 2b, three attention schemes are used in sequence to improve detection accuracy, and their corresponding structures are shown in Figure 2c. Figure 2d illustrates the specific application of DyHead in the detection framework.

The feature pyramid output consists of L distinct levels, which are scaled by the number of levels as $F\in R^{L\times H\times W\times C}$. The variables H, W, and C represent the height, width, and number of channels, respectively, of the mid-level features. It is important to use subject-specific vocabulary when it conveys the meaning more precisely than a similar non-technical term. To obtain the 3D tensor definition $F\in R^{L\times S\times C}$, *S* is further defined as S=H×W. When combined with attention, the general equation for the above tensor is:(1)W(F)=π(F)·F

The attention function is denoted by π(·) and is typically encrypted by a completely contiguous layer. However, training attention functions directly on high-dimensional tensors becomes increasingly expensive in terms of computation as the depth of the network increases. In this way, DyHead splits the function of attention into three separate parts, as shown in Figure 2b, each of which focuses on a single perspective. Single-object attention is shown in Equation (Equation 2):(2)$$W\left(F\right)=\pi_{C}\left(\pi_{S}\left(\pi_{L}\left(F\right)\cdot F\right)\cdot F\right)\cdot F$$

Three functions act on L, S, and C: $\pi_{L}\left(\cdot \right)$, $\pi_{S}\left(\cdot \right)$, and $\pi_{C}\left(\cdot \right)$, respectively. Firstly, the fusion scale-dependent type of attention, which fuses together different scales of semantic information is illustrated in Equation (Equation 3).
(3)$$\pi_{L}\left(F\right)\cdot F=\sigma \left(f\left(\frac{1}{SC}\sum_{S,C}F\right)\right)\cdot F$$
where $\sigma \left(x\right)=\max\left(0,\min\left(1,\frac{x+1}{2}\right)\right)$ is a strictly S-shaped function, and f(·) is a straight-line approximation of a 1 × 1 convolution.

Second, the spatial perceptual awareness module is decomposed into two steps by considering the $\pi_{S}\left(\cdot \right)$ of the high tensor dimension. As shown in Equation (Equation 4), the process of image aggregation is performed across layers at the same spatial location using deformable convolutional sparse attention learning:(4)$$\pi_{S}\left(F\right)\cdot F=\frac{1}{L}\sum_{l=1}^{L}\sum_{k=1}^{K}\omega_{l,k}\cdot F\left(l;p_{k}+▵p_{k};c\right)\cdot ▵m_{k}$$
in which *K* is the sparse sampling location number, $p_{k}+▵p_{k}$ is the self-learning spatial offset, $▵p_{k}$ is the positional offset to the discriminant region, and $▵m_{k}$ is the $p_{k}$ self-learned significance scale of the item. Both are trained using feature inputs at the intermediate level of *F*.

Eventually, task-specific attention comes into play. This allows the function to be dynamically switched on and off according to the task at hand. The principle is shown in Equation (Equation 5).
(5)$$\pi_{C}\left(F\right)\cdot F=max\left(\alpha^{1}\left(F\right)\cdot F_{C}+\beta^{1}\left(F\right),\alpha^{2}\left(F\right)\cdot F_{C}+\beta^{2}\left(F\right)\right)$$
with $F_{C}$ being the *C* function slice, and $\left[\alpha^{1},\;\alpha^{2},\;\beta^{1},\;\beta^{2}\right]T=\theta \left(\cdot \right)$ is a hyperfunction that is taught to check the boundaries of the θ(·) triggering function. The implementation is similar to what is done with adaptive ReLU in that it initially implements pooling of global averages in L × S dimensions to decrease the level of dimensionality and subsequently uses a normalisation layer, two completely related layers, and a shifted S-shaped discrete function to normalise the result to [−1,1].

In the present work, the initial detection head was changed to a DyHead, and the channel count of the predicted output was set to 128. These enhancements enable the sensor head to collect more EC target information, resulting in more accurate detection of the EC target.

### 3.3. ResNet-ACmix Module and AC-E-ELAN Module

#### 3.3.1. ACmix

Both automatic attention and convolution rely highly on the 1 × 1 convolution operation, as found by Pan et al. [36]. To solve this challenge, they designed a fusion model, ACmix, that is an elegant combination of self-detection and convolutional learning.

The first kernel is the convolution [37]: let there be a regular kernel conjugation $K\in R^{C_{out}\times C_{in}\times k\times k}$, products $F\in R^{C_{in}\times H\times W}$ and $G\in R^{C_{out}\times H\times W}$, respectively, as feature input and output matrices, *k* as the core magnitude, and $C_{in}$ and $C_{out}$ as entrance and exit channels, respectively. *H* and *W* represent heights and widths, and $f_{ij}\in R^{C_{in}}$ and $g_{ij}\in R^{C_{out}}$ represent the feature tensor pixels according to *F* and *G*(i,j), respectively. The default convolution is given by Equation (Equation 6).
(6)$$g_{i,j}=\sum_{p,q}K_{p,q}f_{i+p-\left[k/2\right],j+q-\left[k/2\right]}$$

Among them, $K_{p,q}\in R^{C_{out}\times C_{in}}$ is the mass of the nucleus at the (p,q) position, and $p,q\in \left\{0,1,\cdots ,k-1\right\}$.

The second is self-attention [38]: assuming a standard N-headed self-attention module, the attention capacity of the module is calculated as
(7)$$g_{ij}=\prod_{l=1}^{N}\left(\sum_{a,b\in E_{k}\left(i,j\right)}A\left(W_{q}^{\left(l\right)}f_{ij},W_{k}^{\left(l\right)}f_{ab}\right)W_{v}^{\left(l\right)}f_{ab}\right)$$
with ∏ of *N* the concatenated outputs from the individual attention heads, and $W_{q}^{\left(l\right)},W_{k}^{\left(l\right)},W_{v}^{\left(l\right)}$ the projection matrix of queries, keys, and values. $E_{k}\left(i,j\right)$ indicates a pixel-localized area having dimension *k* and centred on (i,j). $A(W_{q}^{\left(l\right)}f_{ij},W_{k}^{\left(l\right)}f_{ab})$ is the projection matrix corresponding to $E_{k}\left(i,j\right)$, which are the attention weights of the internal features. The weights t $A(W_{q}^{\left(l\right)}f_{ij},W_{k}^{\left(l\right)}f_{ab})$ are calculated as follows for the widely used self-attention module:(8)$$A\left(W_{q}^{\left(l\right)}f_{ij},W_{k}^{\left(l\right)}f_{ab}\right)={softmax}_{E_{k}\left(i,j\right)}\left(\frac{{{(W}_{q}^{\left(l\right)}f_{ij})}^{T}{(W}_{k}^{\left(l\right)}f_{ab})}{\sqrt{d}}\right)$$
where *d* is the characteristic dimension of $W_{q}^{\left(l\right)}f_{ij}$.

ACmix reuses intermediate feature mappings in the convolution and self-tuning paths for subsequent aggregations and performs a single 1 × 1 convolution projection on the input feature mappings. Two scalars that can be learned control the intensity of these outputs.
(9)$$F_{out}=ffF_{att}+fiF_{conv}$$

Taking advantage of the best of both worlds, the ACmix module reduces the repetition of highly complex display operations by revealing a strong association between self-attention and convolution. Consequently, it requires less computation than self-attention or convolution alone.

#### 3.3.2. ResNet-ACmix Module

The consistency of the acquired characteristic data is effectively maintained by the introduction of the ResNet-ACmix module in the YOLOv7 backbone module. As shown in Figure 3, the 3 × 3 convolution is replaced by ACmix, which is based on the pinch point structure of ResNet [39] to achieve tunable focusing on various areas and to acquire better significant features. To avoid loss of information and reduce the number of parameters and calculations, the inputs are divided into main and residual inputs. ResNet-ACmix makes the learning of results more sensitive to varying network weights and allows the network to reach greater depths without losing the gradient.

#### 3.3.3. AC-E-ELAN Module

In YOLOv7, E-ELAN [40] employs expanding, shuffling, and combining bases to continually improve the network’s learning capability, resulting in enhanced parameter utilisation and computational performance without compromising the underlying flux gradient. E-ELAN’s feature extraction module is refined by integrating the rest of the RepVgg architecture (i.e., 1 × 1 convolutional branching and hopping branching). However, the intricate network architecture of E-ELAN necessitates a longer training period and more precise hyperparameter tuning, thereby increasing the computational and memory requirements. For this reason, we have designed AC-E-ELAN to replace the E-ELAN module in YOLOv7. The structure of AC-E-ELAN is shown in Figure 4. The AC-E-ELAN structure embeds ACmix modules, which consist of three consecutive 3 × 3 convolutional conjugate blocks, interconnected by branching connections with additional 1 × 1 convolutional conjugate patterns. This facilitates the exchange of information along multiple paths and endows the network with the capacity for rich multimodal learning, thereby enabling the capture of a wider range of feature representations during training. Furthermore, the architecture permits a seamless transition from a composite training mode to a single-path inference process, thereby ensuring efficient and rapid inference performance. Consequently, the AC-E-ELAN architecture not only enhances the model representation but also strikes a balance between computational efficiency and practicality.

### 3.4. SPPFCSPC Module

In the original YOLOv7 network, the spatial pyramid pooling connected spatial pyramid convolution (SPPCSPC) module captures and differentiates between targets of different sizes by performing maximal pooling operations at four different scales. The SPPCSPC module uses three convolutional kernels of different sizes, e.g., 5 × 5, 9 × 9, and 13 × 13. However, this structure increases the amount of network computation by using different sized convolutional kernels for parallel pooling operations. Therefore, we use the spatial pyramid pooling fast connected spatial pyramid convolution (SPPFCSPC) module for pooling operations. As shown in Figure 5, this module replaces the three convolutional kernels of the parallel maximum pooling operation in the original SPPCSPC structure with the serial maximum pooling operation of three identical convolutional kernels. This replacement improves the speed while keeping the sensory field unchanged.

### 3.5. WIoU-Soft-NMS

The bounding box loss function for the YOLOv7 network is the CIoU [41] loss model. This loss function is primarily used for the regression of the predicted frame in order to bring the predicted frame of the object nearer to the position of the object labelled as the ground truth boundary box. It calculates the IoU score of the predicting boundary box and the ground truth boundary box as the ratio of the area of their intersection to the entire area. But this traditional approach can sometimes lead to suboptimal outcomes: for instance, smaller entities are given lower weight in the IoU computation due to their smaller number of pixels, which may cause some bias in looking at these tiny entities.

Wise IoU [42] modifies the IoU loss function to address the problem encountered when training the model. It also incorporates a dynamic tuner that adjusts for object size, occlusion, and background complexity. Particularly in difficult situations, such as smaller objects or objects in complicated surroundings, this dynamic adjustment enables more accurate target acquisition. There are three variants of WIoU: WIoUv1 computes the bounding box loss due to awareness, and WIoUv2 and WIoUv3 compute the bounding box loss by constructing the gradient gain and attaching the focusing mechanism on top of it using algorithmic methods. WIoU-v1 can be solved by Equations (Equation 10) and (Equation 11) as follows:(10)$$L_{WIoUv1}=R_{WIoU}L_{IoU}$$
(11)$$R_{WIoU}=exp\left(\frac{{\left(x-x_{gt}\right)}^{2}+{\left(y-y_{gt}\right)}^{2}}{{\left(W_{g}^{2}+H_{g}^{2}\right)}^{*}}\right)$$
where $W_{g}$ and $H_{g}$ are the minimum closed box dimensions. Similarly, the variables $x_{gt}$ and $y_{gt}$ represent the ground truth box size and height, respectively, while the prediction box size and height are denoted by *x* and *y*, respectively.

Equation (Equation 12) expresses WIoU-v2 through the monotonously focusable factor of WIoU-v1:(12)$$L_{WIoUv2}={\left(\frac{L_{IoU}^{*}}{L_{IoU}}\right)}^{\gamma }L_{WIoUv1}$$

Furthermore, WIoU-v3 utilises a dynamically nonmonotonic focalization procedure that identifies anomalies to characterize the anchor box grade, as shown in Equation (Equation 13).
(13)$$\beta =\frac{L_{IoU}^{*}}{L_{\alpha IoU}}\in \left[0,+\infty \right)$$

To eliminate unnecessarily predicted frames, NMS is commonly used as a post-processing method within a target recognition algorithm. The steps are the following: Enter all potential forecast boundary prediction values $=\;[\left[X_{max},\;X_{min},\;Y_{max},\;Y_{min},\;score\right],\left[*\right],\left[*\right]]$ and a defined IoU limit. The predictive frames filtered by the NMS algorithm are output, i.e., $\left[X_{max},\;X_{min},\;Y_{max},\;Y_{min},\;score\right]$. The non-maximum suppression (NMS) technique retains only the forecast frames with a reliability higher than the upper limit. One significant downside of the NMS approach is that, in the presence of object clutter, the reliability of remaining objects is marginally reduced, while the predictive frames corresponding to other cluttered objects are discarded, severely compromising the ability to detect cluttered objects.

Soft-NMS [43], based on NMS, addresses this issue by adjusting the conventional NMS mechanism with reduced probability for scoring in overlapping regions. The Soft-NMS algorithm follows these steps:First, sort all detected objects from top to bottom.Then, take the top-scoring item and work out the amount of overlap with the other items using the IOUs. Adjust the rest of the objects’ weights accordingly.In the third step, the raw score weighted with the updated weighting is used to compute the reliability of the bounding box.Finally, for each object, do Steps 2 and 3.

Rather than directly removing the M boxes that overlap the largest box larger than a certain threshold, the Soft-NMS algorithm decreases their probability. In this manner, more boxes are preserved and some overlap is avoided.

### 3.6. Improved Network Structure EC-YOLOv7

The architecture of the EC-YOLOv7 network is shown in Figure 6, and the main modules are described. The head detection in DyHead utilizes an attention-based mechanism, which improves the network’s capability to effectively extract significant and profound features. The AC-E-ELAN structure is introduced to improve the initial E-ELAN fabric in YOLOv7 in the proposed EC-YOLOv7 model. We replace the 3 × 3 convolutional boxes with 3 × 3 ACmixBlocks and add branch connectors and 1 × 1 convolutional structures among the ACmixBlocks. This improvement enhances the model’s ability to focus on the useful content and locations of the incoming image patterns, enriches the network’s extracted features, and reduces the model’s training time. In addition, the ResNet-ACmix block is incorporated into the backbone module, behind the CBS in the fourth layer, at the underlying level of the backbone. It effectively retains acquired features and extracts characteristic information from targets with low and complicated backgrounds while accelerating network convergence speed and improving acquisition precision. To enhance the network’s computational speed, we upgraded the neck’s SPPCSPC module to an SPPFCSPC module. WIoU-Soft-NMS is employed in the prediction stage to address the issue of overlapping objects in detection.

## 4. Experiments and Discussion

### 4.1. Datasets

To solve the dataset building problem, we use the publicly available “FICS-PCB [44]” as a dataset. The “FICS-PCB” dataset contains images of printed circuit boards with different component types and under different viewing conditions and is designed to help researchers evaluate performance in difficult real-world scenarios. The dataset comprises 9912 images from 31 PCBs that were either purchased online or extracted from different devices, such as hard disk drive controllers, audio amplifiers, and monitors. The labelling information in the dataset is very comprehensive as it includes virtually all the devices on the PCBs.

We use random cropping to obtain sub-images of size 640 × 640 pixels. To prevent overfitting, it is necessary to apply data augmentation techniques to generate diverse instances. We used conventional geometric transformations and morphologic manipulations such as rotating, flipping, transposing, adding noise, and chromatic transformations involving RGB shifting and arbitrary luminance enhancement. Finally, a total of 1563 images were produced, with the training, validation, and test sets distributed in a ratio of 8:1:1. Table 1 displays the number of training, validation, and test images as well as the input size. The statistical information for the names and numbers of labels after processing is shown in Table 2. Figure 7 displays an extract of our data and depicts numerous ECs densely arranged on a PCB.

### 4.2. Experimental Conditions

Table 3 displays the firmware and hardware settings of the algorithm. During the experiments, colour transformations and noise were added to enlarge the training samples in order to improve the sample balance and, at the same time, to increase the model’s ability to generalize. The batch size is 12, the image pixel size is 640 × 640, and the original training rate is 0.01.

### 4.3. Evaluation Metrics

In the experiments, the mean value of mAP when the IOU threshold is 0.5 (mAP@0.5), the mean value of mAP at different IOU thresholds (from 0.5 to 0.95; step-size 0.05) (mAP@0.5:0.95), and the rate of images per second handled by the module (frames per second, FPS) were taken as evaluation indices. Therein, FPS reacts to the model’s processing speed, and the mAP@0.5 is the geometric mean of the precision achieved when the IoU limit is set at 0.5. For the calculation of mAP, precision and recall are calculated in the following way:(14)$$P=\frac{TP}{TP+FP},\;R=\frac{TP}{TP+FN}$$
(15)$$AP=\int_{0}^{1}P\left(R\right)dR,\;mAP=\sum_{i=1}^{C}\frac{{AP}_{i}}{C}$$
where: TP (true positive) denotes the number of objects classified as positive and predicted as positive that are actually positive samples; FP (false positive) denotes the number of actual negative samples classified as positive and predicted to be positive, i.e., negative samples are tested as positive ones; and FN (false negative) denotes the number of samples in which a target that is actually a positive sample is categorized as a negative sample and at the same time is predicted to be a negative sample. Recall is the percentage of positive specimens in the assay set that are correctly identified as targets. The PR curve displays accuracy on the vertical axis and recall on the horizontal axis and illustrates the interaction of the model’s accuracy at detecting positive samples and its capacity to include every positive sample. AP represents the region under the PR graph, with high levels indicating better classifier performance.

The IoU can compare the degree of overlap between any two graphics to determine the similarity and is defined as the formula shows:(16)$$IoU\;=\;\frac{|A\cap B|}{|A\cup B|}$$
where *A* is the ground truth and *B* is the predicted box.

FPS is the number of frames per second that the model can capture and is calculated as follows:(17)$$FPS=\frac{N}{T}$$
where *N* is how many frames and *T* is how long it takes.

### 4.4. Experimental Results

For validation of the EC-YOLOv7 model, we conducted comparative and ablative trials and compared the outcomes in depth using pre-defined evaluation metrics.

#### 4.4.1. Comparison of Baseline Networks

To assess the efficiency of the YOLOv7 model, we select three network structures in the YOLOv7 model hierarchy for comparison: YOLOv7-tiny, YOLOv7, and YOLOv7x. Table 4 shows that YOLOv7-tiny has fewer parameters, FLOPs, and higher FPS than other models. However, its accuracy is significantly lower and does not satisfy the requirements of the test. YOLOv7x uses more computational power and has a slower inference speed than YOLOv7, although it has a similar accuracy to YOLOv7. Having compared these, we chose YOLOv7 as our reference network to be improved upon.

#### 4.4.2. Performance Comparison of the Proposed Method with Other Networks

To demonstrate the superior performance of the suggested EC-YOLOv7 model, it underwent learning and training on our dataset, and its evaluation metrics, such as mean average precision (mAP@0.5), were compared to those of the popular Faster-RCNN, SSD, RetinaNet, YOLOv5, and YOLOv7 target detection models. Table 5 shows the comparison results. The mAP@0.5 of the EC-YOLOv7 model is 4.9% higher than that of YOLOv7 and higher than that of Faster-RCNN, SSD, YOLOv5, and RetinaNet. This indicates that the EC-YOLOv7 model outperforms other detection algorithms.

Moreover, the performance of our model is benchmarked against that of the improved YOLOv3 model proposed by Li et al. [12]. Our model demonstrates a mAP@0.5 that is 8.7% higher than that of the YOLOv3 model, and the EC-YOLOv7 model exhibits superior speed and a smaller size.

#### 4.4.3. Comparison of DyHead Detection Heads to Other Detection Heads

A commonly used technique to improve the detection performance is to embed the attention mechanism in the target detection model’s detection head. This paper examines the effectiveness of the dynamic head in a YOLOv7 model by incorporating popular attention schemes—a squeeze-and-excitation (SE) module, a convolutional block attention module (CBAM), and an efficient channel attention (ECA) module—into the detection head of YOLOv7. This is shown in Table 6. Figure 8 shows the mAP@0.5 metric curve for the attention.

As can be seen from Figure 8, DyHead demonstrates superior generalisation, robustness and detection efficiency compared to alternative attention sensing heads. Meanwhile, compared with other attention mechanisms, DyHead adds a relatively small number of parameters and has a negligible impact on the model’s complexity.

#### 4.4.4. Experimental Analysis of the Loss Function

The detection performance of three different versions of the loss function—SIoU, DIoU, and WIoU—was evaluated for the WIoU-Soft-NMS method used in this experiment. We then chose the IoU loss function that was most appropriate for the PCB dataset while eliminating the influence of the other components. As demonstrated in Table 3, the precision of the WIoUv2 version is 1.8%, 1.2%, 1.4%, and 1.2% higher than that of the other versions of IoU in Table 7, respectively, and the accuracy of our model when using WIoUv2 is better.

This was followed by tests to assess the sensitivity of Soft-NMS to different IoU loss functions. For YOLOv7 (base) and WIoU, Soft-NMS training results are 0.5% better than before, as shown in Table 8. However, while the accuracy of individual categories decreases by approximately 0.2%, the overall detection performance improves. It is important to note that SIoU and DioU are not compatible with Soft-NMS linking, resulting in a performance decrease of around 1%.

#### 4.4.5. Ablation Study

Ablation experiments were conducted to improve the testing and validation of the model’s performance. Building upon the original YOLOv7 algorithm, we implemented a step-by-step approach to verify the effectiveness of each individual improvement method. Four experimental runs were carried out to test the models, and the outputs are shown in Table 9 and Figure 9.

Table 9 shows that use of the ResNet-ACmix module increased the mAP@0.5 value by 1.1%. The most significant improvement, however, was achieved by using the AC-E-ELAN module as the backbone of the model to capture additional beneficial features, which increased the model’s mAP@0.5 value by 1.8% on top of the benefits from the incorporation of ResNet-ACmix. The speed increase was improved by using the SPPCSPS module. Finally, adding DyHead and using WIoU-Soft-NMS improved the mAP50@0.5 by 1.4% and 0.6%, respectively, compared to the highest results in previous experiments.

#### 4.4.6. Comparison of EC Detection Visualization

In order to evaluate the performance improvement of YOLOv7 after the optimisation of each module, an exhaustive visual comparative analysis was implemented. The results of this analysis are reflected in Figure 10. The original YOLOv7 network (depicted in (d)) is susceptible to limitations to detection accuracy, particularly in complex scenarios with overlapping targets. This results in low detection accuracy and an inability to effectively deal with overlapping targets, which may lead to target misses. The incorporation of the ACmix and AC-E-ELAN modules into the YOLOv7 architecture ((b) and (c)) resulted in a notable reduction in false detections, indicating that these two structural enhancements have a positive effect on improving target differentiation and dealing with occlusion. Furthermore, the integration of the DyHead module not only demonstrates a direct improvement in recognition accuracy (b) but also emphasises the importance of implementing a detection head that incorporates an attentional mechanism for enhancing model performance. Finally, (a) demonstrates the considerable enhancement to the model’s capacity to process highly overlapping targets following the integration of the WIoU-Soft-NMS strategy in YOLOv7. This strategy effectively addresses the issues of false and missed detections. The combination of all the optimisation measures described above, including the incorporation of ACmix and AC-E-ELAN, the integration of DyHead, and the implementation of WIoU-Soft-NMS, has led to a notable enhancement in the robustness, accuracy, and capacity of YOLOv7 to handle complex scenarios in target detection. This reflects the substantial impact of EC-YOLOv7’s improvements to the underlying network.

Analysing all the aspects above, the detection performance of EC-YOLOv7 is better. We selected a series of test datasets containing smaller targets and denser scenarios to verify the effectiveness of individual models in real-world scenarios. The results of the detection are displayed in Figure 11. For better understanding, the false predictions in Figure 11 are pointed out with red boxes.

Figure 11 shows that the proposed method has a reasonable recognition rate in crowded scenes and intricate backgrounds. Comparing (a), (b), and (c) shows that SSD and RetinaNet detect targets incorrectly or miss targets in complicated backgrounds. In addition, a comparison between (a), (d), (e), and (f) shows that YOLOv5, YOLOv7, and Faster RCNN fail to recognise targets in dense scenes. A comparison of (a) and (f) also shows that our model is able to overcome this problem in the case of high EC overlap, and there is no leakage of detection. The superior ability to detect in complex and dense environments is clearly demonstrated by these results. Due to its enhanced performance, our model is a valuable tool for EC detection and recovery and outperforms existing models.

## 5. Conclusions

As electronic manufacturing techniques, materials, and technologies have improved, electronic components (ECs) have become more diverse in shape and size. A key issue in the intelligent manufacturing of electronic products is the image-based recognition of electronic components on PCBs. Currently, various algorithms for detecting PCB electronic components using deep learning have low detection efficiency and low accuracy. Research on how to identify a wide variety of ECs on PCBs using powerful and rapid detection techniques can open up more opportunities for electronic product defect detection, recycling of used electronic components, and robot assembly.

In this work, we propose an extended YOLOv7-based model, called EC-YOLOv7, to detect and classify a wide variety of electrical components on PCBs. To achieve this goal, we use 1563 PCB photos obtained from the “FICS-PCB” dataset and preprocess them to obtain a rich set of EC instances. Based on the original YOLOv7, the DyHead attention-based mechanism is used as the detection head in order to fully detect the feature information of ECs at different scales. In addition, the AC-E-ELAN block is proposed based on the YOLOv7 framework, with the aim of highlighting the goal features. Meanwhile, branch connections as well as 1 × 1 convolutional structures are combined in the ACmixBlock, which increases the calculation performance and storage consumption. For more efficient extraction of deep features during training of the network, the ResNet-ACmix engine is advanced. The SPPFCSPC module is introduced to change the picture merging method of serial channels to parallel channels. Eventually, by using WIoU-Soft-NMS, we overcome the problems of overlapping objects, large localization error, and detection leakage in the detection procedure, and we enhance the generalization ability in the system. As a result, mAP improved from 89.5% to 94.4% compared to original YOLOv7. Overall, the results demonstrate that our approach provides greater speed and precision in detecting ECs in PCBs over conventional object detection methods. EC-YOLOv7 offers a new design for quicker and more precise EC identification.

## Figures and Tables

**Figure 1 sensors-24-04363-f001:**
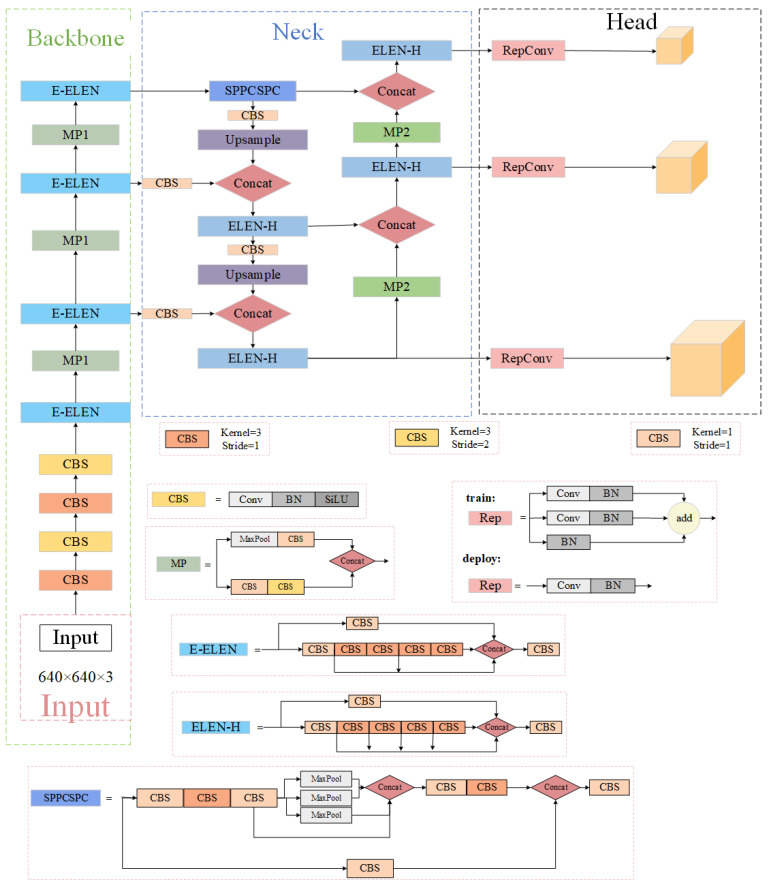
The network structure of YOLOv7.

**Figure 2 sensors-24-04363-f002:**
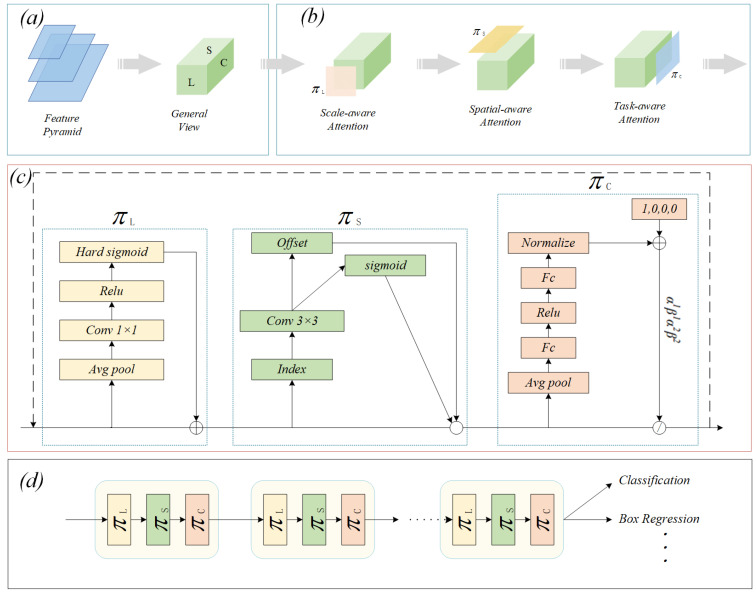
DyHead: (**a**) The input feature tensor of DyHead. (**b**) An illustration of the DyHead approach. (**c**) Details of a DyHead block. (**d**) Specific application of DyHead.

**Figure 3 sensors-24-04363-f003:**
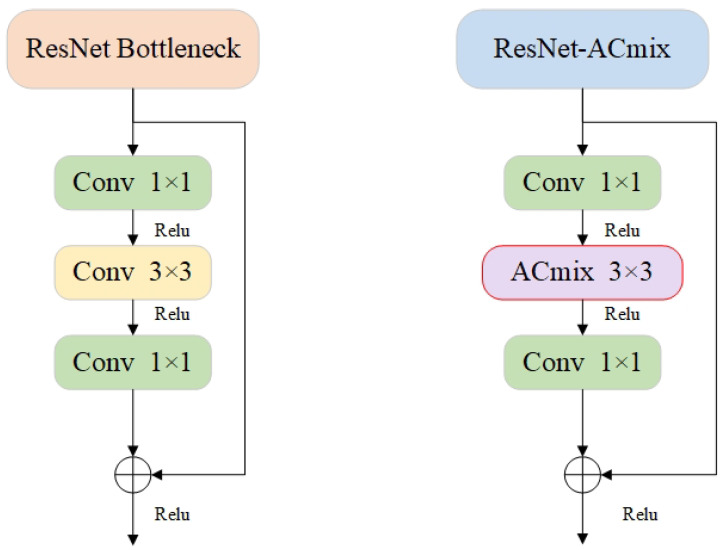
Structure diagram of ResNet-Acmix module. (**left**: ResNet; **right**: ResNet-Acmix).

**Figure 4 sensors-24-04363-f004:**
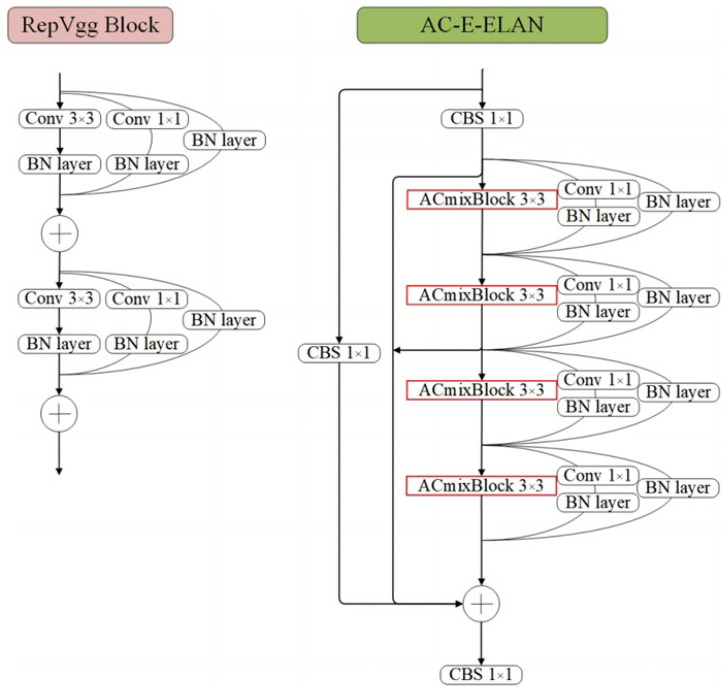
Structure diagram of AC-E-ELAN module. (**left**: RepVgg; **right**: AC-E-ELAN).

**Figure 5 sensors-24-04363-f005:**
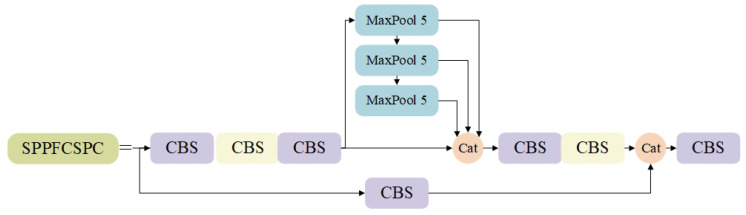
SPPFCSPC module structure diagram.

**Figure 6 sensors-24-04363-f006:**
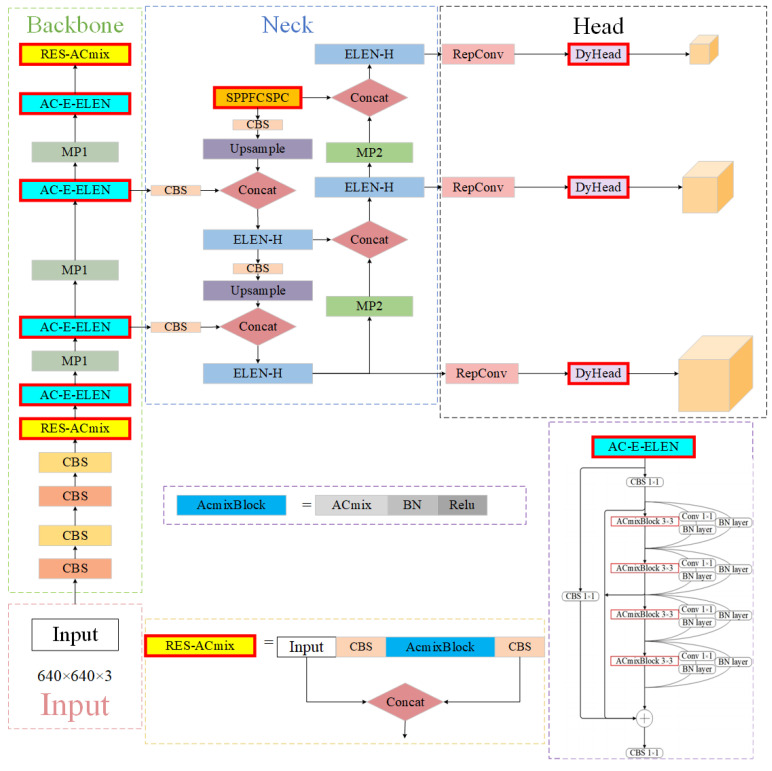
Structure of EC-YOLOv7 model.

**Figure 7 sensors-24-04363-f007:**
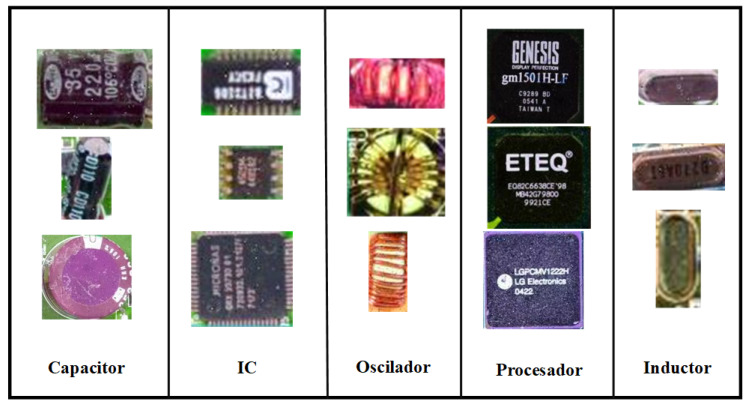
Examples of five types of annotated components.

**Figure 8 sensors-24-04363-f008:**
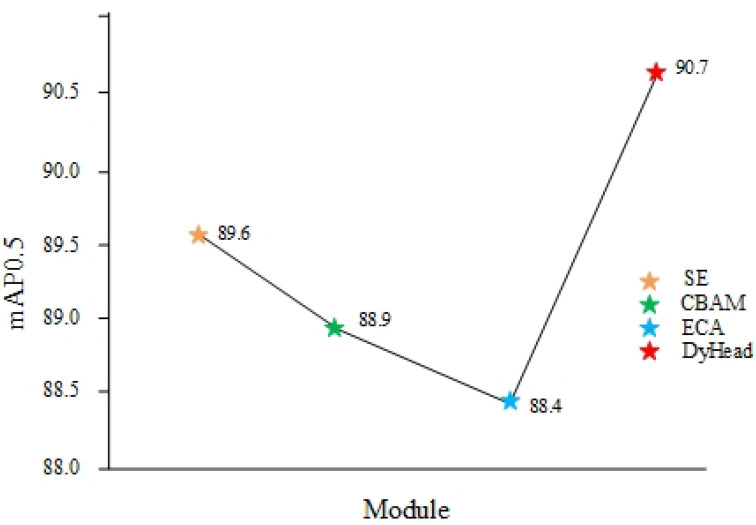
Comparison chart for ablation experiments.

**Figure 9 sensors-24-04363-f009:**
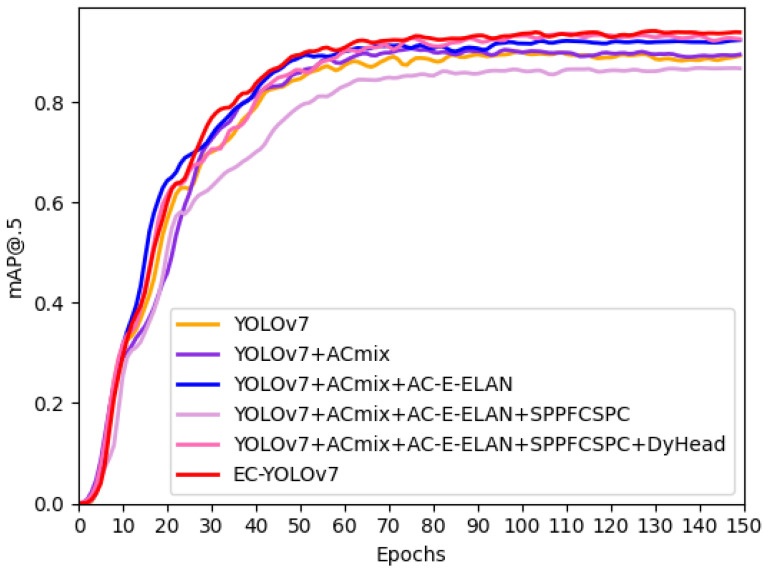
Comparison chart of results of ablation experiments.

**Figure 10 sensors-24-04363-f010:**
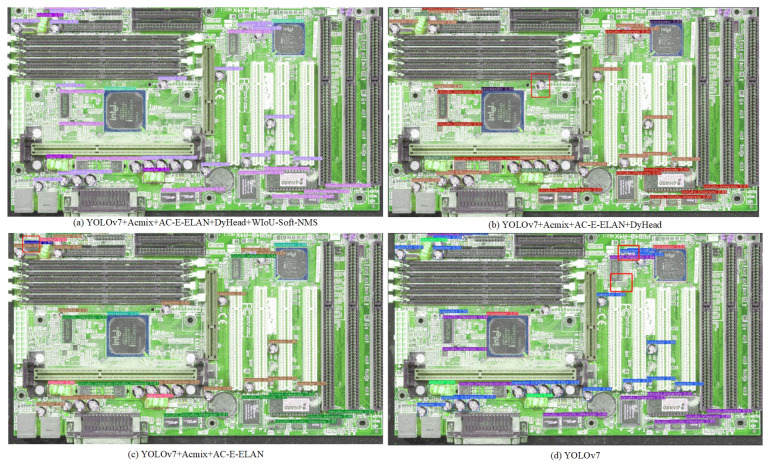
Visual comparison of the detection effect of each improved model.

**Figure 11 sensors-24-04363-f011:**
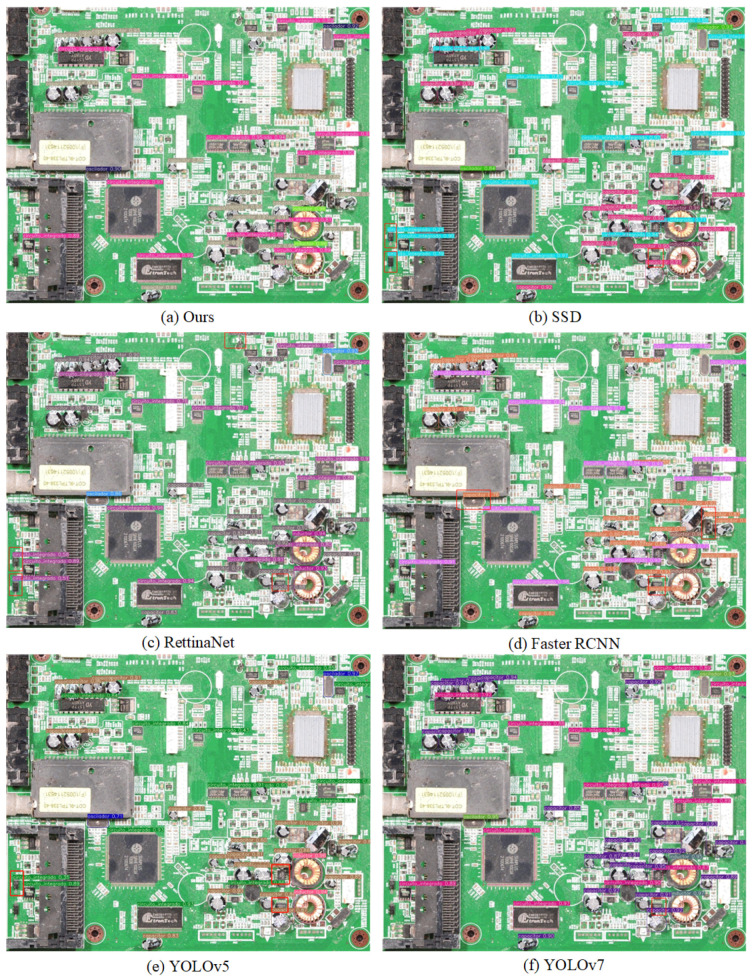
Visual comparison of the detection effects of six models.

**Table 1 sensors-24-04363-t001:** Number of images in each set.

Dataset	Size
train	1251
test	156
valid	156
summary	1563

**Table 2 sensors-24-04363-t002:** Details of the dataset.

Class	Size
capacitor	14,100
IC	8500
oscillator	660
processor	650
inductor	300

**Table 3 sensors-24-04363-t003:** Experimental environment configuration.

Experimental Environment	Configuration Information
Operating system	Windows 11, 64-bit
CPU	Intel® Core™ i7-11700 CPU@2.5GH
GPU	NVIDIA GeForce RTX 3060
GPU acceleration	CUDA 11.6, cuDNN 8.3
Deep learning framework	PyTorch 1.12.0
Scripting language	Python 3.7

**Table 4 sensors-24-04363-t004:** Performance comparison of different models of YOLOv7 network.

Method	Parameters (M)	FLOPs (G)	mAP@0.5 (%)	mAP@0.5:0.95 (%)	FPS (frame/s)
YOLOv7-tiny	6.0	13.1	84.8	17.8	60.9
YOLOv7x	70.8	188.1	88.8	23.8	33.6
YOLOv7	36.5	103.2	89.5	23.7	58.1

**Table 5 sensors-24-04363-t005:** Performance comparison of the popular detection models.

Method	Parameters (M)	FLOPs (G)	mAP@0.5 (%)	mAP@0.5:0.95 (%)	FPS (frame/s)
Faster-RCNN	7.6	169.7	65.1	53.2	30.9
SSD	23.6	188.9	76.1	62.9	50.6
RetinaNet	36.7	104.2	68.5	55.9	69.2
YOLOv5	36.5	103.2	74.2	60.7	47.2
YOLOv7	36.5	103.2	89.5	23.7	58.1
YOLOv3 (Li et al. [26])	61.5	193.9	85.7	47.3	54.6
Ours	36.2	45.6	94.4	63.5	82.5

**Table 6 sensors-24-04363-t006:** Comparison of effects of different attention mechanisms.

Attention Module	Parameters (M)	FLOPs (G)	mAP@0.5 (%)	mAP@0.5:0.95 (%)	FPS (frame/s)
YOLOv7 + SE	36.7	103.4	89.6	12.5	35.5
YOLOv7 + CBAM	34.3	101.9	88.9	23.5	31.6
YOLOv7 + ECA	36.9	104.5	88.4	26.6	33.9
YOLOv7 + DyHead	36.2	98.5	90.7	29.5	59.2

**Table 7 sensors-24-04363-t007:** Comparison of experimental results using different loss functions.

Method	mAP@0.5	mAP@0.5:0.95	Class (mAP@0.5)				
			Capacitor	IC	Oscillator	Processor	Inductor
YOLOv7 (base)	89.5	23.7	95.2	89.9	93.9	75.1	93.4
YOLOv7 + SIOU	89.4	21.6	95.3	89.5	93.1	75.0	94.2
YOLOv7 + DIOU	90.0	22.5	95.3	90.2	94.2	76.0	94.2
YOLOv7 + WIOUv1	89.8	22.1	94.8	89.8	94.1	76.0	94.2
YOLOv7 + WIOUv2	91.2	22.8	95.6	90.6	94.2	81.8	94.0
YOLOv7 + WIOUv3	90.0	22.5	95.0	90.0	91.5	79.7	93.6

**Table 8 sensors-24-04363-t008:** Results of ablation experiments with different IoU loss functions and Soft-NMS.

Method	mAP@0.5	mAP@0.5:0.95	Class (mAP@0.5)				
			Capacitor	IC	Oscillator	Processor	Inductor
YOLOv7 (base) + Soft-NMS	90.9	61.4	95.1	89.7	93.6	82.4	93.5
YOLOv7 + Soft-NMS (SIOU)	88.6	60.9	92.9	88.6	93.6	81.2	93.1
YOLOv7 + Soft-NMS (DIOU)	89.1	61.3	94.1	89.3	95.4	79.4	93.2
YOLOv7 + Soft-NMS (WIOU)	92.2	63.0	95.5	92.7	95.2	83.5	94.3

**Table 9 sensors-24-04363-t009:** The results of the ablation experiments.

Model	ResNet-ACmix	AC-E-ELAN	SPPFCSPC	DyHead	WIoU-Soft-NMS	mAP@0.5	FPS
YOLOv7	×	×	×	×	×	89.5	58.1
	✔	×	×	×	×	90.6	57.6
	✔	✔	×	×	×	92.4	55.2
	✔	✔	✔	×	×	86.6	89.3
	✔	✔	✔	✔	×	93.8	87.2
	✔	✔	✔	✔	✔	94.4	82.5

## Data Availability

Data are contained within the article.

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
