# Peer review of "EC-YOLO: Improved YOLOv7 Model for PCB Electronic Component Detection"

_sensors, 2024, doi:10.3390/s24134363_

Round 1

Reviewer 1 Report

Comments and Suggestions for Authors

1.      Is there an error in σ (x) of formula 3? It is inconsistent with the corresponding formula in reference [35].

2.      In the third paragraph of section 4.3, it is written earlier that "IoUs above the set threshold are recorded as TPs.", those below as FPs.”, In the following formula 16, IOU is calculated based on TP and FP, is there a contradiction between the two?

3.      In the related work, some YOLO based electronic components (EC) detection networks are mentioned. Why is the proposed EC-YOLOv7 not compared with these networks in the experimental section of 4.4.2?

4.      What model is used for the experiment in section 4.4.3 ? Is it based on YOLOv7?

5.      Is YOLOv7 in Table 5 and YOLOv7 (base) in Table 7 under the same experimental conditions, and why are they different[ mAP@0.5 ]? The indicators are the same, while the mAP@0.5 is different?

6.      In the conclusion, it is claimed that "by using WIoU Soft NMS, we overcome the problems of overlapping objects" , how about the experimental results?

Comments on the Quality of English Language

1.      The article contains a large number of grammar errors, many sentence structures are chaotic, meaning is not expressed clearly, and many professional nouns are expressed incorrectly. For example, at the end of section 3.3.1 of the article, "self attention and convolution" is written as "self awareness and connection".

2.      There are many symbol errors in the article. For example, in the second paragraph of page 7, the symbols in "F R ^ (L × H × W × C)" are all capitalized, and the lowercase letters "h, w, c" are used to explain the meaning of the symbols. In formula 5, α and β are superscripted, but the subscript is used to explain the meaning.

Reviewer 2 Report

Comments and Suggestions for Authors

1.The author should compare more advanced object detection algorithms, such as YOLOv8,YOLOv9 and  YOLOv1,etc..

2.A deeper comparative analysis with existing models, especially those that are not based on YOLO architectures, could further validate the effectiveness of the model. 

3.The description of the integrated features, particularly the DyHead detection module,AC-E-ELAN and SPPCSPC module, could benefit from greater detail;

4. It is necessary to provide more visual comparison of test results, especially to highlight the optimization effect of each module.

Comments on the Quality of English Language

A native English review should be done and submitted!

Round 2

Reviewer 2 Report

Comments and Suggestions for Authors

I have no other comments.

Comments on the Quality of English Language

Minor editing of English language required.